# Antioxidant and Anti-Inflammatory Effects of Carotenoids in Mood Disorders: An Overview

**DOI:** 10.3390/antiox12030676

**Published:** 2023-03-09

**Authors:** Paweł Rasmus, Elżbieta Kozłowska

**Affiliations:** 1Department of Medical Psychology, Faculty of Health Sciences, Medical University of Lodz, 90-131 Lodz, Poland; 2Department of Microbiology and Experimental Immunology, MOLecoLAB: Lodz Centre of Molecular Studies on Civilisation Diseases, Medical University of Lodz, 92-215 Lodz, Poland

**Keywords:** depression, carotenoids, mental health, nutrition

## Abstract

Depression has a multifactorial etiology comprising family history and unemployment. This review aims to summarize the evidence available for the antioxidant and anti-inflammatory effects of carotenoids in mood disorders. This review article’s methodologies were based on a search of the PubMed database for all linked published papers. Epidemiological studies indicate that a diet rich in vegetables, fruits, nuts, fish, and olive oil may prevent the development of depression. Antioxidant supplementation has been found to combat various stress-induced psychiatric disorders, including depression and anxiety. A growing body of evidence indicates that carotenoids have both antioxidant and anti-inflammatory. Studies also suggest that poor dietary intake, particularly low intakes of fruit and vegetables and high intakes of fast food and other convenience foods, may increase the risk of developing depression. Thus, dietary interventions have the potential to help mitigate the risk of mental health decline in both the general population and those with mood disorders. Considering that carotenoids have both antioxidant and anti-inflammatory effects, it is expected that they might exert a promising antidepressant effect. Nevertheless, further studies (including interventional and mechanistic studies) assessing the effect of carotenoids on preventing and alleviating depression symptoms are needed.

## 1. Introduction

Some of the most common noncommunicable illnesses are psychiatric disorders such as depression; together, these are responsible for 14.3% of global fatalities. It is estimated that 322 million individuals live with depression globally and that depression is involved in 50–70% of suicides [1]. The World Health Organization predicts depression will become the second most prevalent disorder after ischemic heart disease [2]. Depression is most commonly manifested as hopelessness, disturbances in sleep or appetite, and avoiding social activity, and it also acts as an independent predictor of the beginning of somatic illness [3]. Studies have shown that people with depression have a higher risk of developing lifestyle disorders such as cardiovascular disease, obesity, and diabetes, as well as cancer; it has also been associated with cognitive impairment [4,5,6,7,8]. However, while it is impossible to avoid some of the most common risk factors, such as family history, poverty, and unemployment [9,10], it is possible to modify others, such as a sedentary lifestyle [11], cigarette smoking [12] and alcohol consumption [13].

Depression has a multifactorial etiology comprising functional deficits in monoamine, such as serotonin, and disturbances in the brain’s hypothalamic–pituitary–adrenal (HPA) axis [14]. In addition, emotional disorders such as anxiety and depression are often associated with a reduction in the activity of antioxidant enzymes in the HPA axis. HPA axis hyperactivity is often observed in cases of depression; this correlates with an increase in glucocorticoid concentration, resulting in the development of oxidative stress and glutamatergic excitotoxicity and thus the death of nerve cells, especially in the hippocampus, the key region related to the regulation of mood [15,16,17,18,19,20,21]. Also, chronic psychological stress, known to contribute to the development of depression, causes structural changes in the hippocampus associated with the regulation of cognition [22].

It should be emphasized that although there are many theories of depression development, its accurate pathomechanism is still not fully known. Depression is widely believed to be strongly influenced by immuno-inflammatory mechanisms [23,24,25]. Patients with depression tend to demonstrate significantly elevated levels of various chemokines (CCL2, CXCL10), and pro-inflammatory cytokines, such as interleukin (IL)−1β, IL-6, and tumor necrosis factor (TNF) [26,27,28,29,30]. Some studies found patients with depression to have increased levels of C reactive protein (CRP), an important inflammatory marker [31,32]. Current literature indicates a strong two-way association between the development of inflammation and psychiatric disorders, including depression. It should also be borne in mind that depression is related to an exacerbation of behaviors associated with the development of inflammation, such as nutritional deficiencies/poor eating habits, and addiction to psychoactive substances, such as alcohol, drugs, or smoking [33]. In addition, there is evidence that commensal gut microorganisms, which comprise the gut microbiota, may play an important role in the etiopathogenesis of depression via the gut-brain axis [34]. The gut-brain axis is a two-way communication pathway between the central nervous system and the gut. This communication occurs via hormonal, neurological, and immunological signaling systems, as well as gut microbe metabolites, which trigger changes in neurotransmission, neuroinflammation, and behavior [35,36]. Disturbances in the composition, quality, and functioning of the intestinal microbiota (intestinal dysbiosis) are correlated with some neuropsychiatric disorders, especially depression. Numerous studies have investigated the potential impact of gut microbiota on the onset of depression [37,38,39,40,41,42,43,44,45].

This review aims to summarize the evidence available for the antioxidant and anti-inflammatory effects of carotenoids in mood disorders. This review article’s methodologies were based on a search of the PubMed database for all linked published papers. Various combinations of the keywords, among others, “carotenoids”, “nutrition”, “major depressive disorder”, “MDD”, “unipolar depression”, “mood disorders”, “diet”, “stress”, “oxidative stress”, “antioxidants”, “gut microbiota”, “microbiota”, and “neurodegenerative disorders” were used to make the selection. The World Health Organization and the Centers for Disease Control and Prevention provided the statistical data on the prevalence of mental disorders used in this research.

## 2. The Influence of Diet on the Development and Course of Mood Disorders

A number of studies indicate that an improperly-balanced diet is one of the elements associated with the development of depression and anxiety [46,47]. Recent years have seen a growth in interest in the relationship between nutrients and depression, particularly folic acid, vitamin D, and magnesium [48,49,50]. Mikkelsen et al. (2016) demonstrated a relationship between vitamin B deficiency, including B1, B3, B6, B9, B12, and depression [51,52]. Studies have also shown that a higher level of depression in adolescents is associated with irregular meals [53]. Research indicates that fat-soluble nutrients, such as vitamin E, protect against nerve damage, and low dietary intake is linked to mood changes and depression [54].

It is believed that carotenoid-cleaving enzymes, which take part in the metabolism of carotenoids, play a key role in depression. It should be pointed out that apocarotenoids are formed due to the oxidative breakdown of carotenoids catalyzed by carotenoid oxygenases. Apocarotenoids are, inter alia, retinal, retinol, retinoic acid, and abscisic acid. Some studies describe that retinoic acid, the active form of vitamin A, causes hyperactivity of the hypothalamic-pituitary-adrenal (HPA) axis and leads to the development of typical depressive behaviors. In addition, it has been shown that retinoic acid may cause suicide in some susceptible individuals [55,56].

A cross-sectional study of people of normal BMI by Nguyen et al. (2017) found significantly lower intake of β-carotene equivalent and vitamins C, E, B1, B3, B6, B9, and B5 in those with depressive symptoms [54]. It should be noted that vitamin C insufficiency has also been linked to an increased risk of depression symptoms [54]. Additionally, randomized, placebo-controlled clinical trials have found vitamin C to improve mood and lower the severity of depression in patients [57,58]. Interestingly, however, vitamin C supplementation appeared to have no effect on the depression score in these people [59]. A study conducted among elderly Japanese people showed that the consumption of carotene and vitamin C is associated with less severe depression symptoms [60]. Lower carotenoid concentrations may also reflect unhealthy eating patterns associated with overweight and obesity, which have been linked to an increased risk of depression by inflammation or dysregulation of the HPA axis [5,61,62].

Previous studies have shown that some dietary factors, such as fruit and vegetables, fish, dietary fiber, and some macro- and microelements, may play an important, protective role in the development of depression through their antioxidant and/or anti-inflammatory properties [63,64,65,66]. It is important to note that carotenoids are also known for their antioxidant activity and anti-inflammatory properties [67,68]. Additionally, research shows that depression leads to the development of diseases such as cardiovascular diseases, insulin resistance, metabolic syndrome, and obesity [69,70,71]. These data support the hypothesis that inflammation and oxidative stress may be involved in the pathophysiology of this disorder. Considering that carotenoids have both antioxidant and anti-inflammatory effects, it is expected that they may exert an antidepressant effect.

## 3. The Role of Stress in Unipolar Mood Disorder Pathology

The results of animal studies indicate that psychological stress can increase the level of lipid peroxidation, a significant source of the cell damage caused by reactive oxygen species (ROS) [72], and impair antioxidant protection in the plasma [73,74]. Due to its high oxygen consumption and relatively weak antioxidant defense, the brain is particularly susceptible to oxidative damage, which may increase the likelihood of developing depressive episodes. Therefore, oxidative stress, caused by an imbalance between antioxidants and prooxidants, may play a key role in the remission and chronic course of depressive disorder [75,76].

Milaneschi et al. (2012) report that antioxidants have a beneficial effect on markers of inflammation; their data, obtained in the InCHIANTI prospective population study among seniors in Tuscany, Italy, indicates that inflammatory markers, namely serum levels of the interleukin-1 receptor antagonist (IL-1ra), partially mediate the relationship between carotenoid levels and the development of depressive mood after six-year follow-up [77]. In addition, increased serum carotenoid was associated with a lower risk of developing depression, and this effect was found to be a little distorted by IL-1ra [74]. This relationship between IL-1ra and depression was also confirmed in a meta-analysis by Howren et al. (2009) [78]. Interestingly, studies in an animal model of depression indicated that nuclear factor kappa B is a major mediator, linking stress-induced increases in IL-1β with compromised hippocampal neurogenesis and depressive behavior [79]. The discovery that inflammation mediated the link between plasma carotenoids and depression may indicate that they share the same molecular mechanism. However, carotenoid levels may potentially influence symptoms of depression through various other molecular and behavioral mechanisms. Patients with major depressive disorder (MDD) have been found to demonstrate higher serum levels of nitric oxide (NO) and reactive oxygen species (ROS) than patients without [79,80]. However, the mechanism underlying the relationship between depression symptoms and oxidative stress is not yet understood. So far, studies have shown that different types of depression can be associated with both an overactive (melancholic depression) and an underactive (atypical depression) HPA axis [18]. There is evidence that the HPA axis is associated with an increase in ROS formation; this mechanism has also been shown to involve granulocytes, with significantly higher numbers being observed in depressed patients than in healthy subjects [81,82].

Patients with depression have been found to have a significantly lower mean intake of α-carotene compared to healthy subjects [83]. In addition, depression has been associated with lowered antioxidant levels, as evidenced by low levels of carotenoids and antioxidant enzymes [76,77,84,85]. Black et al. (2016) found reduced levels of carotenoids such as zeaxanthin/lutein, β-cryptoxanthin, lycopene, α-carotene, and β-carotene to be associated with an increase in depression symptoms. Most importantly, this relationship persisted after controlling for diet quality; as carotenoids are only acquired through the dietary route, diet could be considered a significant confounder [3]. Beydoun et al. (2013) report that among the studied carotenoids, β-carotene, lutein, and zeaxanthin levels were inversely related to the incidence of depressive symptoms among US adults [85]. Interestingly, these studies suggest that common genetic factors may influence the relationship between low carotenoid levels and depression: the presence of SNPs associated with low β-cryptoxanthin levels may also influence the occurrence of depression [86].

Some authors also suggest that antidepressants have antioxidant capabilities and may work by lowering pro-inflammatory cytokine levels and ROS generation while increasing those of antioxidants such as superoxide dismutase (SOD) [87,88,89,90,91,92]. Classically-used antidepressants, i.e., selective serotonin reuptake inhibitors, demonstrate an antidepressant effect within two to four weeks of therapy; however, these drugs have common side effects, including cognitive, sexual, and sleep disturbances. Therefore, there is a need to develop new therapeutic strategies that exhibit fewer potential side effects. Interestingly, herbal preparations show promising effects in treating mood disorders without the side effects of synthetic drug use. Jiang et al. (2017) and Sharma et al. (2017) report that treatment with astaxanthin and a combination of lutein and zeaxanthin showed an antidepressant-like effect with the participation of the serotoninergic system [93,94]. Kim et al. indicate that β-carotene has antidepressant properties, which probably result from their influence on reducing the levels of the pro-inflammatory cytokine, i.e., IL-6 and TNF, and increasing the level of brain-derived neurotrophic factor (BDNF) in the brain [95].

In turn, Tsubi et al. [72] and Nouri et al. (2020) [96] found no correlation between the serum level of lycopene and depressive symptoms. Zhang et al. (2016) report that seven days of pretreatment with 60 mg/kg lycopene could reverse LPS-induced depressive behavior in mice based on the tail suspension test and the forced swim test [97]. In turn, a mechanistic study by Lin et al. (2014) found that three-day treatment with 10 mg/kg lycopene reverses the LPS-induced increase in serum TNF and IL-6 concentrations and IL-1β levels in the hippocampus [98]; in addition, pretreatment with 5, 10, or 20 M lycopene inhibited LPS-induced production of cyclooxygenase-2, inducible nitric oxide synthase, and IL-6 in primary cultured microglia via the activation of heme oxygenase-1 [98]. There is a possibility that lycopene supplementation may help maintain cellular homeostasis by restoring normal cell cytokines levels turn. These results suggest that inhibiting neuroinflammation may be a key factor in the antidepressant effects of lycopene.

## 4. Oxidative Stress and Antioxidants in the Course of Mood Disorders

Oxidative stress occurs as a result of an imbalance between the build-up of reactive oxygen species (ROS) or reactive nitrogen species (RNS) and their removal. ROS levels are believed to increase due to various environmental features such as tobacco smoke, ionizing, UV radiation, and by the initiation of cell receptors [99]. At least 5% of inhaled oxygen is converted to ROS, which naturally occurs as a byproduct of aerobic metabolism. In metabolic processes, cytochrome oxidase completely reduces most of the molecular oxygen to water in the mitochondria. Only partially reduced oxygen can react with long-chain molecules such as proteins, carbohydrates, lipids, and DNA. In higher organisms, RNS are produced by the oxidation of one of the terminal guanidonitrogen atoms of L-arginine [100] by nitric oxide synthase. NO can then be converted to various other forms of RNS [101].

The human body has a range of antioxidant defense mechanisms in place to protect against the potentially damaging effects of such active species., for example, by removing free radicals from the body. It is now known that oxidative stress, as well as ROS and RNS, negatively affect a number of cellular processes. When ROS exposure (or generation) increases or antioxidant levels fall, lipids, proteins, and DNA can be damaged, resulting in cell malfunction and even cell death [102]. Importantly, ROS participate in a number of physiological reactions of the body, such as the phagocytosis process [103]. Most importantly, the brain is particularly susceptible to oxidative stress because the level of aerobic respiration is high in the brain tissue. Additionally, brain tissue is rich in polyunsaturated fatty acids (PUFAs) that are susceptible to ROS damage [104].

A growing body of data indicates that ROS may also play an essential role in the pathophysiology of various neurological and psychiatric disorders, including mood disorders. Numerous studies have shown that individuals with neuropsychiatric disorders have higher levels of free radicals, lipid peroxides, pro-apoptotic markers, and altered antioxidant defense mechanisms [105,106,107]. A meta-analysis by Black et al. (2014) found oxidative stress to be elevated in people with MDD and/or depressive symptoms [108]. Importantly, oxidative stress is linked to various socio-demographic, health, and lifestyle variables, including socioeconomic status and smoking, which are also linked to depression [5,102,109,110,111,112,113]. Cigarette smoke has been demonstrated to decrease the levels of carotenoids and other antioxidants in human plasma [114]; it has been proposed that smoking may reduce carotenoid concentration by increasing metabolic rate, resulting in greater oxidative stress [115]. Today it is well known that antioxidants defend against the harmful effects of oxidative stress, which is believed to be associated with depression [116,117].

The antioxidant system consists of enzymatic antioxidants such as inter alia glutathione reductase, SOD, and catalase, as well as non-enzymatic forms such as vitamin C and E, N-acetylcysteine, reduced glutathione, flavonoids, and carotenoids. Carotenoids are natural antioxidants that can effectively prevent oxidative damage [67]. There is evidence that antioxidants exert a neuroprotective effect through their ability to repair the central nervous system and prevent oxidative stress-induced neurodegeneration.

The total antioxidant capacity of a diet has been shown to have an inverse relationship with depression, anxiety, and stress [118,119]. Some data suggest that people with depression consume lower levels of antioxidants in the form of fruit and vegetables compared to those without [120]. In addition, data suggest that patients with depression have lower plasma vitamin E and C levels than those without [88,121]. Vitamin E has been found to exert an antidepressant-like effect in depressed animal models, and this has been attributed to it supporting the enzymatic glutathione-based antioxidant defense system in the hippocampus and prefrontal cortex [122]. However, growing evidence suggests that antioxidant treatment has proven unsatisfactory and even damaging in some oxidation-related diseases such as cancer [123,124,125]. While we know that ROS plays a key role in defense against pathogens and intracellular signaling, the perception is that these compounds are harmful to cells. Likewise, antioxidants should not be regarded as purely beneficial agents. A clinical example of this is the finding that β-carotene supplementation in smokers leads to a significant increase in the incidence of lung cancer [126,127].

## 5. Carotenoids and Their Role in the Course of Depression

Carotenoids are fat-soluble color pigments that belong to the tetraterpene family, present in yellow-orange vegetables and fruits [128]. More than 700 carotenoids have been described, with the major forms being lycopene, β-carotene, ASTA, lutein, and zeaxanthin [129,130,131]. In nature, these pigments are found in many bacteria, fungi, and plants. The groups of carotenoids can also be divided into non-provitamin A and provitamin A (e.g., γ-carotene, β-carotene, α-carotene, and β-cryptoxanthin) [132]. The compounds can also be classified by the presence of specific functional groups: xanthophylls containing oxygen as a functional group (e.g., lutein, zeaxanthin), and carotenes containing only the parent hydrocarbon chain without any functional group (e.g., α-carotene, β-carotene, and lycopene) [132] (Figure 1).

Carotenoids cannot be synthesized de novo by humans and can only be acquired through the dietary route. While around 700 carotenoids have been identified, only six are commonly found in the human diet and blood serum: α-carotene, β-carotene, lutein, zeaxanthin, lycopene, and β–cryptoxanthin. In addition, the typical human diet only includes about 40 carotenoids. Many studies show that providing the body with dietary carotenoids is associated with a reduced risk of developing lifestyle diseases, such as cancer, osteoporosis, diabetes, or cataracts, as well as certain infectious diseases, such as HIV infection [128,132,139]. The data also indicate that carotenoids may reduce the risk of developing CVD by lowering blood pressure and inflammatory markers and increasing insulin sensitivity in muscles, the liver, and adipose tissue. Interestingly, carotenoids could modulate the expression of specific genes involved in cell metabolism [140].

Carotenoids have many beneficial effects. They are mainly known for their antioxidant properties as major scavengers of ROS, including single molecular oxygen and peroxide radicals [132]. Recent epidemiological studies show that higher blood α-carotene and lycopene levels are linked to a lower risk of lung cancer, even among smokers [141]. Interestingly, an ever-increasing body of literature indicates that carotenoids may be effective in the treatment of a variety of cancers, e.g., neuroblastoma [142], cervical cancer [143], and prostate cancer [144]. A large number of existing in vitro and in vivo studies have revealed that carotenoids influence a variety of processes related to the body’s immune-inflammatory response. It has been demonstrated that these compounds influence both the cellular (lymphocyte proliferation, phagocytosis, and NK cell cytotoxicity) and humoral mechanisms of immunity (synthesis and secretion of cytokines) [145,146]. It was found that β-carotene can inhibit the upregulation of heme oxygenase 1 expression in human skin fibroblasts (FEK4) exposed to UV-A [128,147]. In turn, β-carotene has been shown to be less effective in preventing lipid peroxidation [147].

There is a growing body of evidence that the antioxidant and anti-inflammatory properties of carotenoids may promote efficient cognitive function [148,149,150,151,152] by increasing neuronal efficiency or stabilizing the lipid-protein bonds in neuronal membranes. Other neuroprotective mechanisms of carotenoids include enhancement of communication between clefts and modulation of the functional properties of synaptic membranes [151,152,153]. It should be stressed that some studies indicate that higher intake of β-carotene may be related to lower prevalence of depression, anxiety, and stress [104]. Epidemiological studies investigating the relationship between diet, carotenoids, and cognitive maintenance have reported that low levels of carotenoids may play a role in cognitive impairment [154,155]. Prohan et al. (2014) found depressed university male students to have a lower β-carotene intake compared to controls [156,157,158]. Increased β-carotene intake may also relieve depression and anxiety symptoms in cases of low blood antioxidant levels. Antioxidant supplementation has been found to resist stress-induced psychiatric disorders such as depression and anxiety [157].

One particularly potent antioxidant among the carotenoids is lycopene, which can trap singlet oxygen and reduce mutagenesis. Some authors have also suggested that lycopene effectively reduces smoke-generated ROS and modulates redox-sensitive target cells [148]. Research on neurodegenerative and psychiatric disorders also suggests that lycopene may have neuroprotective effects on the central nervous system (CNS) [158,159,160,161,162,163]. It has been shown that long-term intake of lycopene reduces the risk of stroke in men and reduces neuronal apoptosis in the case of cerebral ischemia [162,163,164].

Unlike other carotenoids, xanthophylls such as lutein, astaxanthin, and zeaxanthin are orientated within cell membranes by free hydroxyl groups at each end [165]. Lutein is present in high amounts in green plants and leaves such as spinach, kale, and broccoli. It is also the predominant carotenoid in the primate brain: it was present at almost 10–20 times higher levels in the occipital cortex, prefrontal cortex, and cerebellum than its isomer, zeaxanthin [166]. Zeaxanthin and lutein play a number of roles in both plants and humans, including photoprotection and the maintenance of the structural and functional integrity of biological membranes [167]. Numerous studies have shown that zeaxanthin and lutein exhibit antioxidant properties, thereby protecting cells from potential free radical damage [168]. Lutein, zeaxanthin, and other carotenoids can enter the brain from the blood and accumulate in the retinal macula [169,170]. Lutein is known to accumulate in all cortexes and membranes of the brain.

Serum concentrations of β-carotene, β -cryptoxanthin, and α-carotene can be used as biomarkers to predict the concentration of carotenoids in the brain [148,169,170,171]. It has been documented that a low serum lutein level was associated with depression in Alzheimer’s disease patients [77,172]. Previous research has shown that lutein reduces very low and medium-density lipoprotein levels, as well as inflammation and oxidative stress; it also inhibits the progression of atherosclerosis in humans by lowering serum IL-10 concentration [173,174]. Zeaxanthin can effectively scavenge water- and lipid-soluble peroxide radicals.

It should be stressed that carotenoids, as antioxidants, play an important role in counterbalancing the age-related rise in oxidative stress. It has also been shown that with age, the central nervous system becomes increasingly vulnerable to the impact of free radicals; indeed, seniors are much more vulnerable to protein and lipid damage caused by free radicals, resulting in impaired mitochondrial activity and increased free radical production [174]. The inability to counter oxidative stress may result from progressive neuronal deficits but also from neurodegenerative diseases [175,176,177]. Depression has also been found to cause structural and functional changes in some areas of the brain, especially around the hippocampus [178].

Taking the above into account, carotenoids have the potential to play a protective role in depression through various mechanisms. First, pro-inflammatory cytokines such as IL-6 and TNF impair the expression of BDNF, leading to the onset of depression [179]. Additionally, it should be noted that some studies have shown that patients with depression have lower serum levels of BDNF [95,180]. Interestingly, it has been shown that β-carotene and zeaxanthin may reduce the mRNA expression of IL-6 and TNF [181,182]. What is important, carotenoids have been tested for mechanisms in very important drug targets such as MAO or BDNF, known to be closely related to depression through molecular docking studies for possible inhibitory activity. Recent BDNF and carotenoid docking results conducted by Park et al. (2021) indicate the possibility of allosteric activation of BDNF by carotenoids [183]. On this basis, the authors suggested that dietary carotenoids may be used in the treatment of depressive symptoms. Secondly, this organ is prone to oxidative stress due to high oxygen consumption and high levels of lipids in the brain. At the same time, it is believed that the development of depression is closely related to oxidation and an imbalance between pro- and antioxidants. Studies have shown that people with depression have elevated levels of 8-hydroxy-2’-deoxyguanosine, which is considered a marker of oxidative DNA damage [184]. These results indicate that depression appears to be closely related to oxidative stress. Today we know that carotenoids can effectively remove reactive oxygen species as well as other free radicals. Therefore, considering that carotenoids have both antioxidant and anti-inflammatory effects, it is expected that they may exert an antidepressant effect.

## 6. Dietary Carotenoids in Depression

Studies indicate that a varied diet rich in vegetables, fruits, and nuts, as well as the fish and olive oil characteristic of the Mediterranean diet, may protect against depression. A potential association has been reported between carotenoid intake and the risk of depression [83,95,120,156,185,186]. In addition, α- and β-carotene intake was found to be inversely associated with depressive symptoms in women in late midlife in the US [185], and an inverse association was noted between β-carotene intake and the risk of depression in a case-control study among Korean students [83].

Dietary cryptoxanthin intake was found to significantly influence the risk of depression in a cross-sectional study of American seniors; however, no relationship was observed for another four carotenoids used in the study or total carotenoid intake [120]. Research conducted by Ge et al. (2020) in US adults suggests that both carotenoids such as α-carotene, β-carotene, β-cryptoxanthin, lycopene, and lutein with zeaxanthin, and total carotenoid consumption may be inversely proportional to the risk of developing depressive symptoms [186]. An important role in the etiology of depression may be played by the age-related reduction in total xanthophyll and carotenoid levels that takes place in the frontal lobes [187] (Table 1).

Indeed, xanthophylls have been found to account for 66–77% of total carotenoid levels in the human brain [120]. High dietary carotenoid consumption has been associated with less severe depression in one US study [188], as well as in two other cross-sectional studies in seniors [60,67]. However, it is important to mention that a similar analysis in Australia found no significant associations [189]. Similarly, a significant inverse relationship was found between dietary total carotenoids intake and depression in a cross-sectional analysis of 1274 US adults (30–64 y.o.) [188] and in 500 Japanese seniors (65–75 y.o.) [60]. Also, a similar observation was made for dietary β-cryptoxanthin intake among 278 US seniors (>60 y.o.) [120]. It should be noted that senior populations are more likely to demonstrate comorbidities that may influence the findings.

The role of β-cryptoxanthin and its preventive role in the epidemiology of depression is unknown, and there is not enough study on this topic. All carotenoids are believed to exert antioxidant properties that may protect neural tissue and thus prevent depression [190,191,192]; however, among the carotenoids, this effect is restricted to β-cryptoxanthin. In the human body, only the brain preferentially accumulates xanthophylls, particularly β-cryptoxanthin [187,188]. This is significant because structural and functional alterations in the limbic and cortical structures, particularly in the hippocampus, have been associated with the onset of depression [187]. In addition, xanthophylls degrade more slowly than nonpolar carotenoids and are concentrated within lipid bilayer membranes, particularly in the areas most prone to degradation [193].

Hence, higher dietary β-cryptoxanthin intake appears to lower the risk of depression, but prospective studies are needed to confirm this [127]. Interestingly, these results imply that a Mediterranean-style diet could not only reduce the risk of depression among adults but also could ameliorate the inflammation associated with depression in community-dwelling seniors [194]. Dietary interventions hence serve as a cost-effective approach for promoting healthy aging and reducing the incidence of age-related depression (Table 2).

Other studies have examined the effect of vitamin intake on depression with regard to sex or BMI. No such relationship was observed in underweight participants and overweight men; however, in overweight women, all tested vitamins, except for vitamin A1 (i.e., β-carotene), D, and B3, were associated with lower levels of depression, with vitamin B complex deficiency being most closely correlated with elevated depressive symptoms [197]. Another study [198] in seniors found vitamin E to inhibit the development of depressive symptoms in men; however, lycopene demonstrated 100 times greater singlet oxygen-quenching activity than vitamin E [199]. This is hardly surprising, as lycopene is believed to be the most efficient carotenoid antioxidant; more importantly, consumption has no toxic effects [72]. Indeed, tomatoes account for more than 85% of the dietary intake of lycopene among most people [186]. Such a Mediterranean-style diet may have a beneficial effect on the prevention of depressive symptoms; however, the relationship between tomato/lycopene and depression remains poorly studied [141], and no research has been carried out on the effects of a tomato-rich diet on depression in community-dwelling seniors. However, high dietary intake of various vegetables, especially tomato products, has been found to be independently associated with a reduced risk of depression in community-dwelling seniors (>70 y.o.) [200]. In addition, no relationship was observed between the consumption of vegetables and depressive symptoms. More studies have also reported a relationship between the consumption of tomato products and depression than those associated with dietary antioxidants, such as folic acid and vitamin E [141,200,201].

It needs to be highlighted that depression is the most common mental disorder in the course of neurodegenerative diseases, especially in the course of Alzheimer’s disease (AD). Increasing evidence indicates that depression may be a risk factor for the development of AD, and neurodegeneration in the course of AD may also predispose to the development of depression [202]. Importantly, these conditions are associated with dysfunction of certain centers in the brain, neurotransmission imbalance and dysregulation of the HPA axis, decreased BDNF levels, as well as disruption of the mechanisms that regulate neuroplasticity and cell survival [203]. In addition to BDNF, potential strong candidates for biomarkers implicated in the etiology of MDD contribute to cellular damage and impaired neuronal plasticity and neurotransmission in the prefrontal cortex and hippocampus, including IL-6, TNF, and CRP [204]. Interestingly, Lin et al. (2020) showed that in endothelial cells and monocytes, β-cryptoxanthin, lutein, and lycopene suppressed the fructose-induced expression of TNF and IL-1β [205]. On the other hand, β-carotene is associated with reduced levels of TNF and IL-6 and increased levels of BDNF in animal models [95]. Moreover, Nakamura et al. (2022) observed an inverse association between serum lutein and hs-CRP levels in non-smokers [206]. According to currently available clinical data, nutritional status seems to play an important role in the development and progression of neurodegenerative diseases. In patients with AD, the concentrations of β-carotene, lutein, lycopene, and zeaxanthin were found to be significantly lower compared to a control group [207]. Moreover, US middle-aged and older adults’ serum lutein+zeaxanthin was associated with a reduced risk of all-cause dementia [208]. In turn, astaxanthin-treated mice showed slower memory decline and decreased amyloid-beta (Aβ) and tau protein deposition [209].

Many chronic conditions, including mood disorders, are related to the composition and diversity of the gut microbiota. As dietary patterns influence intestinal microbiota, they could also influence health status. Numerous studies have shown that carotenoids can influence intestinal microbiota in various ways [210]. First, research has shown that carotenoids alter the expression of immunoglobulin A, which is a very important factor in bacterial colonization [211]. Studies in an animal model have shown that supplementation with astaxanthin-enriched yeast for 7 days increased the number of IgA antibody-secreting cells in mice [212,213]. In addition, lycopene has been shown to reduce oxidative stress in the intestines, which may have an effect on the number and diversity of microorganisms that form the gut microbiota [214]. Moreover, there is evidence that vitamin A deficiencies may lead to intestinal dysbiosis, increased susceptibility to infections, and/or damage to the gastrointestinal tract [215]. In a cystic fibrosis observational research, β-carotene intake was shown to be associated with a decrease in *Bacteroides* and an increase in the *Firmicutes* population [216]. In turn, research conducted by Schmidt et al. (2021) demonstrated that a carotenoid-rich diet during pregnancy supports intestinal microbiota diversity [217]. Wiese et al. (2019) noted dose-related improvement in gut microbiota profile with enhanced fractions of *Bifidobacterium adolescentis* and *Bifidobacterium longum* in middle-aged subjects with moderate obesity, to which lycopene was applied for 30 days [218]. It should be noted that β-carotene increases the number of intestinal bacteria that produce short-chain fatty acids and increases the synthesis of SCFAs [219]. Ramos et al. (2018) demonstrated that administering a β-carotene-rich oil to the caw rumen altered the amount of SCFA production (change favoring the production of propionate over acetate). It is well recognized that increasing the propionate-to-acetate ratio promotes a decrease in methane synthesis, which reduces the availability of hydrogen to *Archaea* and so avoids the formation of gut dysbiosis [220]. The study also shows that carotenoids protect the mucosa and epithelial gut barrier, as well as tight junction integrity [221]. Additionally, there is evidence that fucoxanthin, a marine carotenoid, affects the regulation of the intestinal microbiota and especially plays a key role in promoting the growth of *Akkermansia muciniphila*, which can repair intestinal mucosa through immune modulation [222,223,224]. It is worth emphasizing that, in contrast to the impact of carotenoids on the gut microbiota, little is known about the microbiota’s effect on carotenoid metabolites [211].

Many chronic conditions, including depressive-like and anxiety-like mood disorders, are associated with the use of carotenoids. This meta-analysis [225] also found that alpha-carotene, beta-carotene, lycopene, lutein, and zeaxanthin, as well as total carotenoid intake, help decrease the risk of depressive-like symptoms. However, beta-cryptoxanthin levels did not approach statistical significance. Despite the observational studies presented, more longitudinal research or clinical trials should be done to identify optimal carotenoid consumption and plasma/serum levels to prevent or treat mental diseases characterized by significant emotional discomfort and dysfunction. Depression-like symptoms such as loss of pleasure and interest in everyday activities, which are frequently accompanied by headaches, dizziness, and other symptoms of different cerebrovascular disorders, can also be modified by carotenoids [226].

Carotenoids’ anti-PTSD-like action is poorly understood. The recent research [227] addressed this by adopting a single extended stress (SPS) strategy to develop PTSD-like behavioral effects in mice. The results showed that 12 days of lycopene treatment at 10 and 20 mg/kg concentrations, but not at 5 mg/kg dose, positively affected the PTSD-like phenotype induced by SPS, including an increase in freezing time in the contextual fear paradigm, a decrease in time and entries in the open arms in the elevated plus maze test, and a decrease in distance and time in the central area of the open field test, without affecting mouse locomotor activity. Lycopene (20 mg/kg, 12 days) supplementation also inhibited the SPS-induced reduction in brain-derived neurotrophic factor (BDNF) levels in mice’s hippocampus and prefrontal cortex. Overall, lycopene’s anti-PTSD-like effects may be associated with its anti-neuroinflammation and anti-oxidative stress activities.

Lycopene also has preventive and/or therapeutic benefits on the central nervous system (CNS) in several disorders, such as Alzheimer’s disease (AD), Parkinson’s disease (PD), Huntington’s disease (HD), cerebral ischemia, epilepsy, and depression. Lycopene also improves mouse cognition and memory skills in a variety of disease circumstances, including diabetes, colchicine exposure, a high-fat diet (HFD), and aging. Moreover, lycopene can protect against neurotoxicity caused by MSG, trimethyltin (TMT), methylmercury (MeHg), tert-butyl hydroperoxide (t-BHP), and cadmium (Cd). Lycopene administration has particular therapeutic effects in some situations, such as ethanol addiction and haloperidol-induced orofacial dyskinesia. In order to have a comprehensive understanding of the function of lycopene in the CNS, the authors outline and discuss lycopene’s pharmacological effects as well as its possible mechanisms in CNS disorder prevention and/or therapy [228].

The results of research on the spice crocin are promising [229]. It is a natural substance with anti-inflammatory and anti-oxidant effects. Unfortunately, little is known regarding crocin’s anti-inflammatory mechanisms in LPS-induced anxiety and depressive-like behaviors. This study aimed to emphasize crocin’s neuroprotective impact against LPS-induced anxiety and depressive-like behaviors in mice. These findings showed that crocin might be used to treat neuroinflammation and depressive-like behaviors caused by LPS. The result was found to be related to NLRP3 inflammasome and NF-B inhibition, which promoted M1 to M2 phenotypic conversion of microglia.

It should be noted at the end that carotenoids included in food may have differing levels of bioactivity and use for human health. Carotenoids’ stability can be modified by food production and storage.

Popular smoothies made from various fruits and vegetables provide vitamins, minerals, and bioactive components such as carotenoids and fiber to the diet. Thermal treatments increased the microbiological shelf-life of a fruit smoothie (mild and intensive). The color assessment was shown to be marginally better for smoothies following ultrasonic treatment rather than heat treatment. Although heat treatment lowered the concentration of carotenoids in foods, it was discovered that liberation, micellarization, and, therefore, bioaccessibility had a positive effect [230].

In another study [231], liquid chromatography coupled with a photodiode array detector was used to monitor changes in the number of carotenoids and their true retentions (% TR) throughout the manufacturing of pumpkin puree, as well as the stability of such compounds after 180 days of storage. Cooking resulted in greater losses than commercial sterilizing. During the preparation and storage of pumpkin puree, high losses of xanthophylls such as lutein and violaxanthin were observed. These losses demonstrate the low stability of these compounds. After processing, the major carotenoids, pro-vitamin A carotenes, namely α-Carotene and all-trans-β-carotene for C, had high retentions (>75%). The puree samples showed a slight degree of isomerization of β-carotene but with low concentrations of cis-isomers. The concentrations of these carotenoids did not change considerably after 180 days of storage.

Even though carotenoids are hydrophobic, their absorption from dietary items is often quite low. This review [232] summarizes current studies on fruits and vegetables that associate specific product characteristics, such as (micro)structural characteristics and the presence or addition of lipids, to carotenoid bioaccessibility. Lipids are naturally present in specific fruits and vegetables, although in small levels. Additionally, they are frequently used in manufacturing fruit and vegetable-derived products (for example, low-fat emulsions), where they play an essential role in satiety control, mouth feel, and taste. Since carotenoids are lipophilic micronutrients, the oil phase acts as a hydrophobic domain in which carotenoids may be dissolved. Therefore, its presence is essential for carotenoids to be incorporated into micelles and then taken up by humans.

Interesting research [233] focused on the characterization of tomato peel extracts. The presence of lycopene and β-carotene was indicated by the chromatographic profile. According to the in silico results, heat treatment of lycopene molecule pairs at temperatures over 100 °C may reduce their biological activity. In addition, the thermal degradation of total carotenoids, β-carotene, and lycopene was studied at temperatures ranging from 100–145 °C, which revealed that the enriched peanut and cotton oils with extract had longer induction time values, indicating a protection factor of tomato-derived carotenoids. The findings emphasized the thermostability of carotenoids in vegetable oils, as well as the possibility of using tomato peel as a valuable source of biologically active compounds with antioxidative properties. The study provides cumulative information for selecting industrial processing parameters from the perspective of preserving bioactives and obtaining the desired quality end products.

## 7. Conclusions

In conclusion, poor dietary habits, i.e., low fruit and vegetable intake and high fast food/convenience food intake, may increase the risk of depression. Carotenoids have the potential to play a protective role in the development of depression through various mechanisms. The most pertinent point is that pro-inflammatory cytokines such as IL-6 and TNF impair the expression of BDNF, which leads to the onset of depression. Recent BDNF and carotenoid docking results indicate the possibility of allosteric activation of BDNF by carotenoids. On this basis, it is suggested that dietary carotenoids may be used in the treatment of depressive symptoms. Another interesting point to consider is the fact that the development of depression is closely related to oxidation and an imbalance between pro- and antioxidants. It is well known that carotenoids can effectively remove reactive oxygen species as well as other free radicals. Therefore, considering that carotenoids have both antioxidant and anti-inflammatory effects, they are expected to exert a promising antidepressant effect. Nevertheless, further studies (including interventional and mechanistic studies) assessing the effect of carotenoids on preventing and alleviating depression symptoms are needed.

## Figures and Tables

**Figure 1 antioxidants-12-00676-f001:**
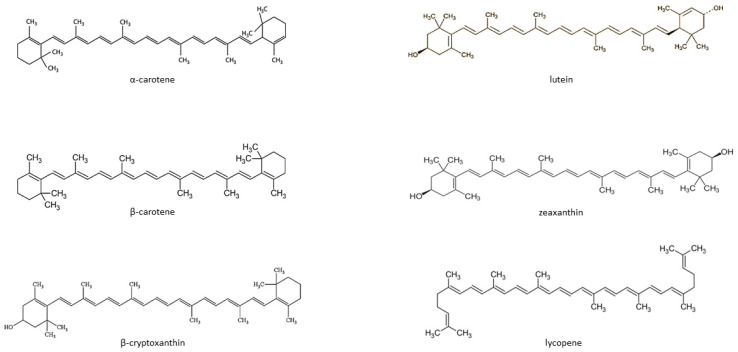
Chemical structure of some common carotenoids [133,134,135,136,137,138]. Source: Own elaboration based on the indicated data.

**Table 1 antioxidants-12-00676-t001:** Localization of certain carotenoids in the brain. Source: Own elaboration based on the indicated data.

Carotenoids	Brain Region	Crossing the Blood-Brain Barrier?	References
α-carotene	Occipital cortex	no	[187]
β-carotene	Cerebellum Frontal cortex Occipital cortex Temporal cortex	no	[169,171,187]
β-cryptoxanthin	Cerebellum Frontal cortex Occipital cortex Temporal cortex	yes	[120,169,171,187]
Lutein	Cerebellum Frontal cortex Occipital cortex Temporal cortex Retina	yes	[166,169,170,171,187]
Zeaxanthin	Cerebellum Frontal cortex Occipital cortex Temporal cortex Retina	yes	[166,169,170,171,187]
Lycopene	Cerebellum Frontal cortex Occipital cortex Temporal cortex	no	[169,171,187]

**Table 2 antioxidants-12-00676-t002:** Dietary sources of carotenoids [195,196]. Source: Own elaboration based on the indicated data.

Carotenoids	Food Sources
α-carotene	butternut squash; collards; tomato (red, ripe); carrot; beans (green); pepper (red, sweet); corn; okra; avocado;
β-carotene	broccoli; grapefruit (red/pink); carrot; asparagus; peas (green); brussels sprouts; mango; okra; zucchini (with skin); tomato (red, ripe); beans (green);
β-cryptoxanthin	papaya; tangerine; orange; watermelon; collards; nectarine; avocado; peach; orange juice; grapefruit (red/pink); mango; pepper (red, sweet);
lutein + zeaxanthin	corn; beans (green); lettuce (iceberg); cabbage; tangerine; orange; tomato (red, ripe); papaya; peach; eggs; melon; kale; watermelon; grapefruit (red/pink)
lycopene	apricots; grapefruit (red/pink); tomato (red, ripe); watermelon; papaya

## Data Availability

Not applicable.

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
