# Peer review of "Antioxidant and Anti-Inflammatory Effects of Carotenoids in Mood Disorders: An Overview"

_antioxidants, 2023, doi:10.3390/antiox12030676_

Round 1
Reviewer 1 Report
Neurodegenerative diseases constitute a major problem of public health that is associated with increased risk of mortality and poor quality of life. Poor dietary habitual behaviors and malnutrition are considered as a major problem that worsens the prognosis of patients suffering from neurodegenerative diseases, especially psychiatric disorders.
In this aspect, the present review is aimed to critically collect and summarize all the available existing clinical data as far as concern the clinical impact of nutritional assessment in neurodegenerative diseases, highlighting on the crucial role of nutritional status in disease progression and management. According to the currently available clinical data, the nutritional status of patients with depressive disorders seems to play a very important role in the development and progression of neurodegenerative diseases. A correct nutritional evaluation of neurodegenerative disease patients and a right nutrition intervention is essential in monitoring their disease.
However, this review only aims to highlight recent possible contributions of carotenoids to alleviating the neuropsychiatric degenerative diseases based on the rising prevalence of depressive disorders. Such possible nutritional status and depressive disorders contributors in disease progression and management did not mention. According to the currently available clinical data, the nutritional status of patients seems to play a very important role in the development and progression of metabolic and neurodegenerative diseases. A recommended nutritional approach or nutrition intervention data may be necessary to reveal or even alleviate in the patients with neurodegenerative disease.
This was a well-reviewed brief article in neuropsychiatric disease, especially in unipolar disease. The mentioned nutritional assessment methods in reviewed papers are mainstream for clinics. For a perspective aspect for disease prevention, some update potential biomarkers (BDNF or others?) should be discussed. Overall, this was a well-organized brief review article.
Author Response
To the Reviewer 1:
Reviewer #1: Neurodegenerative diseases constitute a major problem of public health that is associated with increased risk of mortality and poor quality of life. Poor dietary habitual behaviors and malnutrition are considered as a major problem that worsens the prognosis of patients suffering from neurodegenerative diseases, especially psychiatric disorders.
In this aspect, the present review is aimed to critically collect and summarize all the available existing clinical data as far as concern the clinical impact of nutritional assessment in neurodegenerative diseases, highlighting on the crucial role of nutritional status in disease progression and management. According to the currently available clinical data, the nutritional status of patients with depressive disorders seems to play a very important role in the development and progression of neurodegenerative diseases. A correct nutritional evaluation of neurodegenerative disease patients and a right nutrition intervention is essential in monitoring their disease.
Thank you for your valuable comments, we really appreciate them. We have thoroughly looked through your comments and corrected the manuscript accordingly. Below are our answers to all your comments.
However, this review only aims to highlight recent possible contributions of carotenoids to alleviating the neuropsychiatric degenerative diseases based on the rising prevalence of depressive disorders. Such possible nutritional status and depressive disorders contributors in disease progression and management did not mention. According to the currently available clinical data, the nutritional status of patients seems to play a very important role in the development and progression of metabolic and neurodegenerative diseases. A recommended nutritional approach or nutrition intervention data may be necessary to reveal or even alleviate in the patients with neurodegenerative disease.
This was a well-reviewed brief article in neuropsychiatric disease, especially in unipolar disease. The mentioned nutritional assessment methods in reviewed papers are mainstream for clinics. For a perspective aspect for disease prevention, some update potential biomarkers (BDNF or others?) should be discussed. Overall, this was a well-organized brief review article.
Following your comment, we enriched our review with some important information, among others highlighting that depression is the most common mental disorder in the course of neurodegenerative diseases, especially in the course of Alzheimer's disease (AD). In line with your guidance, in the revised version of our manuscript, we emphasize that there is now increasing evidence that depression may be a risk factor for the development of AD, and neurodegeneration in the course of AD may also predispose to the development of depression and that these conditions are associated with dysfunction of certain centers in the brain, neurotransmission imbalance and dysregulation of the HPA axis, decreased BDNF levels, as well as disruption of the mechanisms that regulate neuroplasticity and cell survival. At the same time, we added some information about potential biomarkers, and their updates under the influence of carotenoid-based nutritional interventions (Please see 6. Dietary Carotenoids in Depression section, line: 426-448).
We sincerely hope that our detailed explanations and corrections made in the text will allow you to re-analyze this manuscript and that the corrected version of our manuscript will be found this time adequate to be published in the Antioxidants
Reviewer 2 Report
Thank you for allowing me to review the article entitled “Antioxidant and Anti-inflammatory effects of in Mood Disorders. An Overview.” (antioxidants-2249641).
The aim of this article is an overview on antioxidant and anti-inflammatory effects of carotenoids in mood disorders.
The abstract presents the interest of the subject and raises the hypothesis, but it should include the main objective, the methodology used, as well as the main results and conclusions of the study. This more complete summary will allow the reader an understanding of the work presented.
The manuscript is structured, which facilitates understanding, and the writing allows easy follow-up of the hypotheses raised and the available results.
It is a review specifically of the type: an overview... as the manuscript indicates in the title. This type of review is a comprehensive review of what has been published in recent years and raises aspects to be developed in future research in the coming years, as indicated by its conclusions. The abstract raises the issue of the multifactorial origin of depression and the role that nutrition can play, specifically the intake of carotenoids, as reflected in the latest publications. and the objective of evaluating this knowledge about intake and depression is proposed to propose future research and interventions. in the introduction the importance of the subject is raised, it is identified that more than 322 million individuals globally have depression with a suicide rate between 50 and 70%, which tells us about the magnitude of the problem that is addressed as well as the different hypotheses that have arisen in different works on factors that intervene in the etiology of depression. Among the factors that are related to the etiology of depression are antioxidants, which seem to intervene in the hypothalamic–pituitary–adrenal axis. Proposing in this way the review carried out. The methodology used for this review has been structured into sections: 2. The influence of diet on the development and course of mood disorders 3. The role of stress in unipolar mood disorder pathology 4. Oxidative stress and antioxidants in course of mood disorders 5. Carotenoids and their role in the course of depression. 6. Dietary Carotenoids in Depression Each of these sections addresses the working hypothesis from a specific perspective, which allows a better understanding of the subject. This structure seems to me very adequate to organize the information. Finally, it presents the conclusions in a practical way and is based on the results identified in the different works in the study sections. Therefore, the conclusions are consistent with the information reviewed.
Author Response
To the Reviewer 2:
Reviewer #2: Thank you for allowing me to review the article entitled “Antioxidant and Anti-inflammatory effects of in Mood Disorders. An Overview.” (antioxidants-2249641).
The aim of this article is an overview on antioxidant and anti-inflammatory effects of carotenoids in mood disorders.
Thank you for your valuable comments, we really appreciate them. We have thoroughly looked through your comments and corrected the manuscript accordingly. Below are our answers to all your comments.
The abstract presents the interest of the subject and raises the hypothesis, but it should include the main objective, the methodology used, as well as the main results and conclusions of the study. This more complete summary will allow the reader an understanding of the work presented.
We agree with your suggestion. Therefore, we rewrote abstract and the new version contains the information you indicated - main objective, the methodology used, as well as the main results and conclusions of the study (please see Abstract section, line: 10-24).
The manuscript is structured, which facilitates understanding, and the writing allows easy follow-up of the hypotheses raised and the available results. It is a review specifically of the type: an overview... as the manuscript indicates in the title. This type of review is a comprehensive review of what has been published in recent years and raises aspects to be developed in future research in the coming years, as indicated by its conclusions. The abstract raises the issue of the multifactorial origin of depression and the role that nutrition can play, specifically the intake of carotenoids, as reflected in the latest publications. and the objective of evaluating this knowledge about intake and depression is proposed to propose future research and interventions. in the introduction the importance of the subject is raised, it is identified that more than 322 million individuals globally have depression with a suicide rate between 50 and 70%, which tells us about the magnitude of the problem that is addressed as well as the different hypotheses that have arisen in different works on factors that intervene in the etiology of depression. Among the factors that are related to the etiology of depression are antioxidants, which seem to intervene in the hypothalamic–pituitary–adrenal axis. Proposing in this way the review carried out. The methodology used for this review has been structured into sections: 2. The influence of diet on the development and course of mood disorders 3. The role of stress in unipolar mood disorder pathology 4. Oxidative stress and antioxidants in course of mood disorders 5. Carotenoids and their role in the course of depression. 6. Dietary Carotenoids in Depression Each of these sections addresses the working hypothesis from a specific perspective, which allows a better understanding of the subject. This structure seems to me very adequate to organize the information. Finally, it presents the conclusions in a practical way and is based on the results identified in the different works in the study sections. Therefore, the conclusions are consistent with the information reviewed.
We sincerely hope that our detailed explanations and corrections made in the text will allow you to re-analyze this manuscript and that the corrected version of our manuscript will be found this time adequate to be published in the Antioxidants.
Reviewer 3 Report
Rasmus and Elzbieta presented antioxidant and anti-inflammatory effects of carotenoids on mood disorders such as depression and anxiety. This is an interesting review paper, introducing BDNF and carotenoid docking to induce allosteric activation of BDNF by carotenoids. Furthermore, the authors discuss the effects of carotenoids on oxidative stress in course of mood disorders. There are, however, several issues to be addressed to further improve the manuscript.
1. Regarding cited papers, 51 papers are within a few years (after 2016), 25% of the total number of 203 papers. The number of references within 5 years needs to be increased a bit more.
2. The effects of carotenoids on depression-like and anxiety-like behaviors should also be discussed.
3. Recently, it is sometimes discussed that the relationship between depression and the brain-gut correlation, particularly with regard to intestinal bacteria. The section on the relationship between carotenoids and gut bacteria is also intriguing.
4. Autoclave sterilization of food is a common food processing technique in modern society to increase shelf life of foods. How are carotenoids affected in autoclaved foods such as retort-packed food other than fast/convenience foods?
Author Response
To the Reviewer 3:
Reviewer #3: Rasmus and Elzbieta presented antioxidant and anti-inflammatory effects of carotenoids on mood disorders such as depression and anxiety. This is an interesting review paper, introducing BDNF and carotenoid docking to induce allosteric activation of BDNF by carotenoids. Furthermore, the authors discuss the effects of carotenoids on oxidative stress in course of mood disorders. There are, however, several issues to be addressed to further improve the manuscript.
Thank you for your valuable comments, we really appreciate them. We have thoroughly looked through your comments and corrected the manuscript accordingly. Below are our answers to all your comments and questions.
- Regarding cited papers, 51 papers are within a few years (after 2016), 25% of the total number of 203 papers. The number of references within 5 years needs to be increased a bit more.
We agree with your suggestion. Therefore, in the revised version of our manuscript we have increased the number of references within 5 years. We would like to pointed out, that in the new version of our paper, we have improved and expanded the Introduction and Dietary carotenoids in depression section, thus the manuscript included new references. In total, we added 44 new references – 36 of them were published in the last 5 years (please see References section, line: 593-624; 994 – 1163
- The effects of carotenoids on depression-like and anxiety-like behaviors should also be discussed.
According to your instructions, in the revised version of our manuscript, we have added information on effects of carotenoids on development of depression-like and anxiety-like disorders. (please see 6. Dietary Carotenoids in Depression section, line: 481-522).
- Recently, it is sometimes discussed that the relationship between depression and the brain-gut correlation, particularly with regard to intestinal bacteria. The section on the relationship between carotenoids and gut bacteria is also intriguing.
Following your comment, in the revised version of our manuscript, we have added information on not only the impact of the gut microbiota on the development of mood disorders, but also enriched the manuscript with information on the impact of carotenoids on the gut microbiota (please see 1. Introduction section, line: 63-73; and 6. Dietary Carotenoids in Depression section, line: 449-480).
- Autoclave sterilization of food is a common food processing technique in modern society to increase shelf life of foods. How are carotenoids affected in autoclaved foods such as retort-packed food other than fast/convenience foods?
At the end we noticed that carotenoids found in food may have various levels of bioactivity and use for human health. Food production and storage can modify the stability of carotenoids. We mentioned information for the selection of industrial processing parameters from the perspective of bioactive preservation and producing the appropriate healthy end products. (please see 6. Dietary Carotenoids in Depression section, line: 423-565).
We sincerely hope that our detailed explanations and corrections made in the text will allow you to re-analyze this manuscript and that the corrected version of our manuscript will be found this time adequate to be published in the Antioxidants.